# Niacin Alleviates Dairy Cow Mastitis by Regulating the GPR109A/AMPK/NRF2 Signaling Pathway

**DOI:** 10.3390/ijms21093321

**Published:** 2020-05-08

**Authors:** Wenjin Guo, Juxiong Liu, Wen Li, He Ma, Qian Gong, Xingchi Kan, Yu Cao, Jianfa Wang, Shoupeng Fu

**Affiliations:** 1College of Veterinary Medicine, Jilin University, Changchun 130062, China; guowj17@mails.jlu.edu.cn (W.G.); juxiong@jlu.edu.cn (J.L.); liwen9918@mails.jlu.edu.cn (W.L.); mahe9916@mails.jlu.edu.cn (H.M.); gongqian17@mails.jlu.edu.cn (Q.G.); kanxc19@mails.jlu.edu.cn (X.K.); yucao18@mails.jlu.edu.cn (Y.C.); 2College of Animal Science and Veterinary Medicine, Heilongjiang Bayi Agricultural University, Daqing 163000, China; wjflw@sina.com

**Keywords:** mastitis, GPR109A, niacin, BMECs, AMPK/NRF-2, autophagy

## Abstract

Mastitis is one of three bovine diseases recognized as a cause of substantial economic losses every year throughout the world. Niacin is an important feed additive that is used extensively for dairy cow nutrition. However, the mechanism by which niacin acts on mastitis is not clear. The aim of this study is to investigate the mechanism of niacin in alleviating the inflammatory response of mammary epithelial cells and in anti-mastitis. Mammary glands, milk, and blood samples were collected from mastitis cows not treated with niacin (*n* = 3) and treated with niacin (30 g/d, *n* = 3) and healthy cows (*n* = 3). The expression of GPR109A, IL-6, IL-1β, and TNF-α in the mammary glands of the dairy cows with mastitis was significantly higher than it was in the glands of the healthy dairy cows. We also conducted animal experiments in vivo by feeding rumen-bypassed niacin. Compared with those in the untreated mastitis group, the somatic cell counts (SCCs) and the expression of IL-6, IL-1β, and TNF-α in the blood and milk were lower. In vitro, we isolated the primary bovine mammary epithelial cells (BMECs) from the mammary glands of the healthy cows. The mRNA levels of *IL-6, IL-1β, TNF-α*, and autophagy-related genes were detected after adding niacin, shRNA, compound C, trans retinoic acid, 3-methyladenine to BMECs. Then GPR109A, AMPK, NRF-2, and autophagy-related proteins were detected by Western blot. We found that niacin can activate GPR109A and phosphorylate AMPK, and promote NRF-2 nuclear import and autophagy to alleviate LPS-induced inflammatory response in BMECs. In summary, we found that niacin can reduce the inflammatory response of BMECs through GPR109A/AMPK/NRF-2/autophagy. We also preliminarily explored the alleviative effect of niacin on mastitis in dairy cows.

## 1. Introduction

Dairy cow mastitis is an inflammatory disease caused by the co-infection of various pathogenic microorganisms. Dairy cow mastitis is a serious problem that hinders the development of the dairy industry worldwide [1]. In recent decades, the problem of dairy cow mastitis has not been well resolved, and it still causes tremendous economic losses to the world’s dairy industry [2,3]. The global economic loss caused by dairy cow mastitis is as high as USD 35 billion annually. The main cause of mastitis in dairy cows is the invasion of pathogenic microorganisms into the mammary glands of dairy cows. Although antibiotics can kill pathogenic microorganisms in dairy cows, there is no good method for treating some drug-resistant strains or the inflammation caused by the infection [4,5]. After bacterial infection, a large number of endotoxins [6,7] are produced in the mammary glands. The main components of these endotoxins are lipopolysaccharides (LPS). It can cause a strong immunogenic response [8]. LPS can also trigger the immune response of bovine mammary epithelial cells (BMECs) or macrophages, thereby releasing a large number of pro-inflammatory factors, damaging the BMECs and leading to mastitis [9,10]. Therefore, in follow-up experiments, we used LPS to construct an inflammation response model of BMECs in vitro. 

Studies have shown that the inflammation of BMECs induced by LPS can reduce milk production and levels of milk fat and milk protein [11,12,13]. The inflammatory response of mammary epithelial cells can also cause damage to the blood–milk barrier. Therefore, it is important to control the inflammation of BMECs in mastitis. However, there are few anti-inflammatory drugs that can be used in dairy cows in the clinic at present. Previous studies have shown that many inflammatory processes are accompanied by oxidative damage; therefore, inducing activation of antioxidant proteins could help alleviate LPS-induced inflammation [14]. Some studies have also shown that inflammation is closely related to autophagy and that enhancing autophagy can lead to alleviated inflammation [15]. Therefore, we speculate that the anti-inflammatory function of GPR109A is related to autophagy.

Previous studies have shown that GPR109A can inhibit the development of colitis [16]. Niacin, a ligand of GPR109A, has also been found to inhibit the inflammation of vascular endothelial cells [17]. These studies suggest that GPR109A may play a role in controlling the inflammatory reaction. How does GPR109A work? After consulting the literature, we found three articles that reported on GPR109A in 2003 [18,19,20], followed by an increasing number of publications suggesting that GPR109A plays an important role in reducing inflammation [21,22]. Previous studies have found that GPR109A is closely related to glycometabolism and inflammation [23]. S Elangovan et al. found that GPR109A inhibits the occurrence of breast tumors by inhibiting cell survival [24]. These studies suggest that GPR109A may alleviate inflammation and activate the innate immune pathway. Therefore, we designed the following experiments to explore the mechanism of niacin inhibition of mastitis through GPR109A.

## 2. Results

### 2.1. Expression of GPR109A and Pro-Inflammatory Factors in Healthy and Mastitis Dairy Cows

To clarify the relationship between GPR109A and mastitis in dairy cows, healthy dairy cows and mastitis dairy cows were selected for experiments designed to detect the expression of GPR109A in mammary glands. A small tissue of mammary gland of healthy cow and mastitis cow was taken out by puncture. First, to examine the reliability of the mammary gland samples, pathological changes and the mRNA levels of pro-inflammatory factors (IL-6, TNF-α, and IL-1β) in the mammary gland were detected by hematoxylin and eosin (H&E) staining or real-time (RT)-PCR, respectively. The H&E staining analysis revealed that a large number of neutrophils had infiltrated the mammary acini of mastitis dairy cows, but neutrophils did not invade the mammary acini of healthy cows (control group; Figure 1a). The H&E staining analysis revealed that a large number of neutrophils had infiltrated the mammary acini of the mastitis dairy cows (Figure 1a) but not those of the healthy dairy cows. Real-time (RT)-PCR showed that, compared with those in the healthy dairy cows, the mRNA levels of IL-6 (Figure 1d), TNF-α (Figure 1e), and IL-1β (Figure 1f) in the mastitis dairy cows increased significantly. Second, the protein and mRNA levels of GPR109A were detected by immunohistochemistry and qRT-PCR three times, respectively. The results showed that, compared with the levels in the healthy dairy cows, both the protein and mRNA levels of GPR109A were significantly upregulated in the mastitis dairy cows. Therefore, GPR109A might play an important role in the development of mastitis.

### 2.2. Activation of GPR109A Inhibits LPS-Induced Inflammatory Response in BMECs

BMECs are the sentinel cells of the mammary gland as they are the first cells to recognize pathogen-associated molecular patterns (such as LPS and LTA) and trigger the inflammatory response. To clarify the effect of GPR109A on the development of mastitis, we examined the effect of GPR109A on the LPS-induced inflammatory response in the BMECs. First, the effect of niacin (GPR109A agonist), a transfection reagent, and GPR109A-shRNA1 on the cytotoxicity of the BMECs was detected by the CCK-8 test. The results showed that niacin, the transfection reagents, and GPR109A-shRNA1 did not cause cytotoxicity (Appendix A). Next, to determine whether GPR109A signaling is required for suppression of the inflammatory response in BMECs, the cells were pre-treated with niacin for 1 h and then stimulated with LPS for 24 h, and the mRNA levels of pro-inflammatory factors (*IL-6, TNF-α*, and *IL-1β*) in the LPS-induced BMECs were examined by real-time (RT)-PCR. The results revealed that niacin significantly attenuated the increased mRNA expression of *IL-6, TNF-α*, and *IL-1β*, but in the cells in which GPR109A was knocked down by GPR109A-shRNA1, this effect was abolished (Figure 2a–c). These results suggested that the activation of GPR109A could inhibit the LPS-induced inflammatory response in the BMECs.

### 2.3. RA and 3-MA Abolish the Anti-Inflammatory Function of GPR109A in LPS-Induced BMECs

NRF-2 signaling and the process of autophagy have anti-inflammatory effects. To examine whether NRF-2 and autophagy were associated with the anti-inflammatory function of GPR109A, BMECs were pre-treated with NRF-2-signalling inhibitor RA or autophagy inhibitor 3-MA before they were incubated with niacin and LPS. The results showed that activation of GPR109A significantly attenuated the increased mRNA expression of *IL-6, TNF-α*, and *IL-1β*, but RA or 3-MA abolished this effect (Figure 2d–i). These results suggested that GPR109A could inhibit the inflammatory response via NRF-2 and autophagy.

### 2.4. Activation of GPR109A Promotes Autophagy via the NRF-2 Signalling Pathway in BMECs

To study whether GPR109A can activate NRF-2, BMECs were treated with niacin for 0, 3, 6, 12, and 24 h, and niacin increased the protein levels of GPR109A, NRF-2, and HO-1 (Figure 3a–d). It is generally believed that niacin can enhance their function by upregulating the expression of receptors. Therefore, after niacin treatment, both GPR109A and NRF-2/HO-1 were upregulated, suggesting that niacin could upregulate NRF-2/HO-1 signaling by activating its receptor GPR109A. To further verify that activation of GPR109A could activate NRF-2, GPR109A was knocked down by shRNA. The results from the Western blot and immunofluorescence assays showed that niacin promoted NRF-2 entry into the nucleus and that GPR109A-shRNA1 abolished this effect, as expected (Figure 3e–i). Further, the results showed that niacin could promote and enhance the transcriptional activity of NRF-2, and GPR109A-shRNA1 abolished this effect (Figure 3j). HO-1 is a target gene of NRF-2, and the protein levels of HO-1 were also examined by Western blot. The results showed that the activation of GPR109A upregulated the protein level of HO-1 (Figure 3e,f). These results suggested that GPR109A activation initiated the NRF-2 signaling pathway in the BMECs. Studies have shown that NRF-2 could promote autophagy by regulating the expression of autophagy-related genes. To verify the effect of NRF-2 on autophagy in BMECs, the BMECs were treated with RA for 6 h, and the expression of autophagy-related genes was determined by qRT-PCR. The results showed that the mRNA levels of ATG12, ATG4D, and p62 did not change significantly after adding RA, but the relative expression of ATG5, ULK1, ATG4B, Beclin, LC3B, and ATG7 decreased significantly (Figure 3k). These results suggested that activation of GPR109A promoted autophagy via the NRF-2 signaling pathway in the BMECs.

### 2.5. Upregulated GPR109A Activates the NRF-2 Signalling Pathway by Regulating AMPK Signalling and the Interaction between P62 and Keap1 in BMECs

Previous studies have shown that AMPK could regulate NRF-2 nuclear import. To further study how GPR109A promotes the NRF-2 movement to the nucleus in BMECs, we examined the phosphorylation level of AMPK and found that niacin could promote the phosphorylation of AMPK (Figure 4a,b) and that GPR109A-shRNA1 abolished this effect. These results indicated that the activation of GPR109A could activate AMPK signaling. Previous studies have shown that the ratio of (AMP+ADP)/ATP is the key signal for activating AMPK. The ATP, ADP, and AMP content in the niacin + GPR109A-shRNA1 group and the GPR109A-shRNA1 group were significantly lower than those in the niacin group (Figure 4f–h). The ratio of (AMP+ADP)/ATP showed a similar trend (Figure 4i), indicating that GPR109A activated AMPK by regulating the ratio of (AMP+ADP)/ATP. Further results showed that CC (an inhibitor of AMPK) inhibited niacin-induced nuclear translocation of NRF-2 (Appendix A). Our study also found that the activation of GPR109A promoted NRF-2 nuclear translocation by increasing the interaction between p62 and Keap-1 (Figure 4e). These results suggested that the activation of GPR109A activates the NRF-2 signaling pathway by regulating AMPK signaling and the interaction between P62 and Keap1 in BMECs.

### 2.6. Activation of GPR109A Enhances the Autophagy in LPS-Induced BMECs

The results illustrate that GPR109A activated NRF-2 to promote autophagy via AMPK or by increasing the interaction of p62 and Keap-1. To further verify the role of autophagy in LPS-induced inflammation in BMECs, the effects of GPR109A on autophagy were detected by Western blotting, electron microscopy, and fluorescence staining. The results from the Western blot analysis revealed that, compared with the LPS group, the LPS + niacin group had significantly increased LC3B protein levels and decreased the P62 levels (Figure 5a–c). Compared with the no-treatment group, the niacin group also had increased LC3B protein levels and decreased P62 protein levels, although the differences were not significant (Figure 5a–c). Transmission electron micrographs showed that there were autophagosomes in the BMECs treated with niacin or LPS + niacin (Figure 5d). In addition, compared with the LPS group, the LPS + niacin group had significantly increased levels of autophagy-related genes P62, ATG4D, Beclin, ATG12, ATG5, and LC3B (Figure 5e–g,j–l). The level of ULK1 was slightly increased after niacin alone was added (Figure 5i), while the level of ATG7 slightly increased after LPS treatment (Figure 5m). The levels of ATG4B did not change significantly (Figure 5h). These results indicated that autophagy could be observed after GPR109A is activated, which may be related to the anti-inflammatory effect of GPR109A.

### 2.7. Activated GPR109A Initiates Signaling through the AMPK/NRF-2/HO-1 Pathway in LPS-Treated BMECs

Previous studies have shown that the MAPK signaling pathway may be associated with the activation of autophagy [25,26,27]. To further study the pathway by which GPR109A activates autophagy, we also examined the MAPK signaling pathway. The results showed that niacin had no significant effect on the MAPK signaling pathway. Compared with the LPS group, the niacin group did not have reduced phosphorylation of P38, JNK1/2, or ERK1/2 (Figure 6a–d), which indicated that the MAPK signaling pathway was not involved in autophagy in the BMEC inflammation model in vitro.

To detect whether GPR109A could activate the AMPK/NRF-2/HO-1 signalling pathway in the BMEC inflammation model in vitro, we detected the protein levels of AMPK, p-AMPK, HO-1, β-tubulin, N-P65, N-NRF-2, and laminin B. We found that the protein levels of N-P65 (Figure 6e,h) were significantly downregulated, while the protein levels of p-AMPK (Figure 6e,f), HO-1 (Figure 6e,g) and N-NRF-2 (Figure 6e,i) were significantly upregulated. Compared with the LPS group, the results of the double luciferase reporter gene test in the LPS + niacin group showed that the activity of pNF-кB was significantly suppressed, while that of NRF-2/ARE was significantly enhanced (Figure 6j,k).

### 2.8. Ameliorating the Effect of Activated GPR109A on Mastitis in Dairy Cows

To study whether GPR109A has an anti-inflammatory role in cow mastitis, the mastitis dairy cows were fed diets with or without niacin for 7 d, and the protein levels of pro-inflammatory factors were detected, and the somatic cells were counted. Compared with the non-niacin group, serum IL-6, TNF-α, and IL-1β in the niacin group were significantly decreased after niacin was fed (Appendix A). Compared with the non-niacin group, there were significant decreases in IL-6, TNF-α, and IL-1β in milk (Appendix A). The somatic cell count (SCC) in milk also decreased significantly compared with the non-niacin group (Appendix A). Compared with the findings from 0 d, the pro-inflammatory factors were downregulated after 7 d in each group, but the anti-inflammatory effect was better on the 7th day after niacin was fed.

## 3. Discussion

We demonstrate that GPR109A plays an important role in anti-mastitis. The main anti-inflammatory mechanism of GPR109A involves the activation of the AMPK signaling pathway to promote NRF-2 nuclear import. NRF-2 entry into the nucleus promotes the expression of HO-1 and autophagy-related genes. Our results show that activation of GPR109A could significantly reduce the expression of N-P65, IL-6, IL-1β, and TNF-α in BMECs, thus inhibiting mastitis. 

Dairy cow mastitis is a very serious mammary disease, and severe inflammation can even cause the death of dairy cows. The main cause of mastitis is bacterial endotoxins, such as LPS [28]. LPS can increase the expression of IL-6, TNF-α, and IL-1β in the mammary glands of dairy cows, which leads to the occurrence and development of mastitis [29]. We detected the expression of pro-inflammatory mediators and GPR109A in the mammary glands of normal and mastitis dairy cows. The results showed that the expression of GPR109A and pro-inflammatory mediators were significantly upregulated, suggesting that GPR109A may be closely related to mastitis.

BMECs are the main group of cells responsible for lactation in dairy cows, so the inflammatory reaction of the mammary epithelial cells in the occurrence of mastitis directly affects the lactation ability and mammary health of dairy cows. Therefore, we isolated and cultured primary BMECs from dairy cows for subsequent experiments. GPR109A belongs to the G protein-coupled receptor family [30], which binds chemicals from the surrounding cellular environment and activates a series of signaling pathways within cells [31,32]. Studies have shown that GPR109A is highly expressed in neutrophils [33], macrophages and adipocytes [34]. Activation of GPR109A significantly inhibits the expression of pro-inflammatory mediators in adipocytes or monocytes. Interestingly, we found that GPR109A was highly expressed in BMECs and during an inflammatory response. Therefore, GPR109A may play an important role in mastitis. Our results suggested that niacin could reduce the expression of IL-6, TNF-α, and IL-1 β in the BMECs. The main ligand of GPR109A is niacin, which indicates that GPR109A may also play an anti-inflammatory role in BMECs. Then, we further studied the specific mechanism of GPR109A in influencing anti-inflammatory functions. Previous studies have shown that GPR109A could alleviate inflammation in enteritis or microglia by inhibiting AKT and NF-κB signaling pathways [21,35]. Other studies have shown that GPR109A can alleviate colitis by mediating treg cells [16]. However, the specific mechanism of GPR109A in anti-mastitis has rarely been reported.

NRF-2 is an important transcription factor involved as an antioxidant during oxidative stress in cells, and HO-1 is a direct downstream target gene of NRF-2. Studies have shown that NRF-2 could promote the transcription and translation of downstream autophagy-related genes, thereby promoting cell autophagy [36]. Many studies have shown that HO-1 and autophagy could play an important anti-inflammatory role in the inflammatory response [37,38]. Studies have also shown that chlorogenic acid could play an anti-inflammatory role through the NRF-2/HO-1 signaling pathway and that activation of NRF-2/HO-1 could inhibit inflammation in monocytes. These studies suggest that the NRF-2/HO-1 signaling pathway plays an important role in the anti-inflammatory process. However, the relationship between GPR109A and NRF-2 has rarely been reported. It was interesting to find that GPR109A could significantly enhance the transcriptional activity of NRF-2 and promote NRF-2 nuclear import. These results suggest that GPR109A could activate the NRF-2/HO-1 signaling pathway and thus play an anti-inflammatory role. At the same time, we further validated the finding that GPR109A activates NRF-2/HO-1 in the BMEC inflammation model. The results showed that GPR109A could also activate the NRF-2/HO-1 signaling pathway in the BMEC inflammation model. These findings suggested that GPR109A could play an anti-inflammatory role through NRF-2/HO-1.

Autophagy was described by Ashford and Porter in 1962 after they discovered that there is a “self-eating” phenomenon among cells. Autophagy fulfills the metabolic needs of cells and promotes the renewal of some organelles. In addition, many reports have shown that NRF-2 promotes the expression of downstream autophagy genes [39]. Moderate autophagy can reduce the inflammatory response of cells. Studies have shown that the NRF-2/HO-1 signaling pathway can enhance autophagy and thus help heal liver ischaemia-reperfusion injury in mice. Interestingly, we also detected the same mechanism in the BMECs. Therefore, we speculate that GPR109A may also play an anti-inflammatory role by activating autophagy.

Some studies have shown that p-AMPK can promote NRF-2 transcription into the nucleus of raw264.7 and play a role in inhibiting inflammation [40]. AMPK, as an energy sensor in cells, is very sensitive to energy changes in cells. Our results also indicated that AMPK phosphorylation is associated with intracellular energy metabolism and that p-AMPK promotes NRF-2 nuclear import. We found that niacin significantly increases ATP, ADP, and AMP levels in BMECs, while downregulated GPR109A significantly suppresses intracellular energy metabolism, as indicated by the (AMP + ADP)/ATP ratio. This finding suggests that GPR109A might promote the phosphorylation of AMPK by altering intracellular energy metabolism; that is, GPR109A might be associated with intracellular energy metabolism. The NRF-2 nuclear import might also be related to the interaction between P62 and Keap-1. Studies have shown that the interaction between Keap-1 and P62 promotes NRF-2 nucleation [41]. Interestingly, our results also showed that GPR109A could enhance the mutual effects of P62 and Keap-1, thus promoting NRF-2 nuclear import.

A large number of studies have shown that mastitis can seriously affect the milk yield and milk quality of dairy cows [42], so alleviating mastitis is an important means to improve the milk yield and quality of dairy cows. In addition, some studies have shown that niacin can significantly increase milk yield and milk protein in dairy cows [43]. Therefore, the study of niacin to alleviate mastitis is of great significance to increase the milk yield and quality of dairy cows. Previous studies have shown that niacin can alleviate oxidative stress caused by inflammation. Wu NC et al. have found that niacin could reduce pulmonary oxidative stress (such as MDA) and lung inflammation (such as TNF-α) [44]. In our experiments, we have confirmed the mechanism of niacin in alleviating mastitis in dairy cows in vitro. To further verify the function of niacin in cow mastitis, we added niacin in dairy cows. Interestingly, we found that niacin significantly increased the level of innate immunity in the mammary glands and decreased the levels of IL-6, TNF- α, and IL-1β in the milk and serum of dairy cows compared with the non-niacin group.

On the basis of the above experiments, we found that niacin might promote autophagy by activating the GPR109A/AMPK/NRF-2 signaling pathway. In addition, niacin can also significantly alleviate mastitis in dairy cows.

## 4. Materials and Methods

### 4.1. Reagents

Trans retinoic acid (RA), LPS, phenylmethanesulfonylfluoride fluoride (PMSF), and niacin were purchased from Sigma (Saint Louis, MO, USA). Compound C (CC) and 3-methyladenine (3-MA) were purchased from Selleck Chem (Shanghai, China).

### 4.2. Antibodies

AMPK (1:1000, CAT: 5832S), p-AMPK (1:1000, CAT: 2535S), ERK1/2 (1:1000, CAT: 4695S), p-ERK1/2 (1:1000, CAT: 9101L), JNK1/2 (1:1000, CAT: 9252S), p-JNK1/2 (1:1000, CAT: 9251L), P38 (1:1000, CAT: 9212S), p-P38 (1:1000, CAT: 4631S) and NF-κB-p65 (1:1000, CAT: 8242S) were purchased from Cell Signaling Technology (Boston, MA, USA), P62 (WB: 1:1000, Co-IP: 1:50, CAT: 66184-2-2g), LC3B (1:1000, CAT: 18725-1-AP) and Keap1 (1:1000, CAT: 10503-2-AP) were purchased from Proteintech (Rosemont, IL, USA), HO-1(1:1000, CAT: GK284419-11), NRF-2 (WB: 1:1000, IF: 1:150, CAT: Gk298097-1) were purchased from Abcam (Cambridge, UK), GPR109A (WB: 1:200, IHC: 1:50, CAT: NBP1-92180) were purchased from Novus Biologicals (Shanghai, China), β-Tubulin (1:6000, CAT: M05613-4) were purchased from Bosterbio (Pleasanton, CA, USA), IgG (Co-IP: 1 μg, CAT:A7028) were purchased from Beyotime (Shanghai, China), (HRP)-conjugated anti-mouse (1:10,000, CAT: BA1051) or anti-rabbit secondary antibodies (1:10,000, CAT: BA1055) were purchased from Bosterbio (CA, USA).

### 4.3. Animals and Experimental Design

The study was conducted at Jilin University, Changchun, China. The animal studies were performed in compliance with the established guidelines (Kilkenny et al. (2010), McGrath and Lilley (2015)). The trial was approved by the Institutional Animal Care and Use Committee of the Jilin University (23 April 2019) and was registered on the Jilin University (SY201904013, 23 April 2019). Lactating dairy cows, with an average body weight (BW) of 622 ± 62 kg, were housed in individual tie stalls inside a barn of a commercial dairy farm in Heilongjiang, China. The cows were fed in the stalls, had ad libitum access to clean water, and were allowed to roam freely after feeding. Screenings for mastitis-positive dairy cows and healthy dairy cows were performed by the California mastitis test (CMT) [45]. Six cows with similar severity of mastitis and three healthy cows were screened by CMT. The dairy cows with mastitis were divided into two groups, three cows in each group. At present, the treatment cycle of mastitis in dairy farms is generally 7–10 days, so 7 days is selected as the treatment cycle in this study, and the dosage is selected according to the previous research [46,47]. Healthy dairy cows as no treatment (control) group. One mastitis group was treated with rumen-bypassed niacin (30 g/d, 7 d) (niacin group), and the other group received only sham gastric perfusion (non-niacin group). We collected the peripheral blood from the tail vein of the dairy cow with a 5-mL syringe and centrifuged it at 3000 rpm for 25 min to separate the serum. We also collected milk through the mammary gland of the dairy cows, each of which collected was about 10 mL, centrifuged at 3000 rpm for 25 min, and separated from the whey. Serum and whey of each group were collected before and after treatment.

### 4.4. Plasmid Construction

The GPR109A knockdown plasmid (GPR109A-shRNA1) was constructed by GenePharma (Suzhou, China). We found that GPR109A-shRNA1 (shRNA) could significantly reduce the expression of GPR109A by qRT-PCR (Appendix A). So we chose GPR109A-shRNA1 in subsequent experiments.

### 4.5. ELISA

The protein levels of TNF-α (Cat. H-80095), IL-6 (Cat. H-80071) and IL-1β (Cat. H-78807) in serum and milk were determined by ELISA kits according to the manufacturer’s instructions (Hengyuan, Shanghai, China).

### 4.6. Cell Culture

Bovine mammary epithelial cells (BMECs) were isolated and cultured from the mammary glands of the dairy cows (Holstein cows). The tissue extracted from the mammary gland of Holstein cows in the middle of lactation was cut into 1-mm small pieces and incubated in the humidified air containing 5% CO2 at 37 ℃. When cells migrate from the tissue covering 80% of the bottom, the tissue is removed. Epithelial cells and fibroblasts were isolated with 0.25% trypsin and 0.15% trypsin plus 0.02% EDTA. The dispersed cells were inoculated on cell culture bottles (Corning) in DMEM (GIBCO). In the basic medium, 5 mg/mL transferrin, 5 mg/mL insulin, 5 mg/mL prolactin, 1 mg/mL hydrocortisone (Sigma), 10 ng/mL epidermal growth factor, 1% glutamine, 1% penicillin, 1% streptomycin, and 10% fetal bovine serum (Clark) were added [48].

### 4.7. Separation of Nucleus and Cytoplasm

We inoculated the BMECs into the cell culture dish. When the cells reached about 80%, GPR109A-shRNA, niacin, or LPS were used to treat the BMECs. Cells were washed twice with cold PBS after the above treatment [49,50] and subjected to nuclear and cytosolic fractionations according to the Nuclear and Cytoplasmic Protein Extraction Kit (Cat: P0027, Beyotime, Shanghai, China). 

### 4.8. Cell Counting Kit-8 Assay

The effect of niacin on cell viability was determined using the CCK-8 assay. The BMECs were treated with niacin (1 and 2 mM) for 24 h. Subsequently, 10 μL of CCK-8 (Saint-Bio, Shanghai, China) was added to each well. After 1 h, the absorbance (OD) was measured at 450 nm in a microplate reader.

### 4.9. Real-Time (RT)-PCR

The total RNA was extracted from the BMECs and mammary glands using TRIzol reagent (Life Technologies, Carlsbad, CA, USA). The qRT-PCR assay was performed in accordance with the experimental methods that our group has previously used [51]. The primer sequence is shown in Table 1.

### 4.10. Reporter Gene Activity of NRF-2/ARE and NF-кB

The BMECs were plated at a density of 3 × 10^4^ cells per well in a 6-well plate and cultured to reach 75% confluence. Some cells were divided into 3 groups: NT, niacin (1 mM), and niacin + GPR109A-shRNA1. Using the transfection reagent (Roche Diagnostics GmbH, Mannheim, Germany), shRNA, pGL4.37 (luc2P/ARE/Hygro vector), and pGL4.74 (hRluc/TKvector) plasmids were transfected into cells. After the treatment of niacin (1 mM), a dual-luciferase reporter assay system (Dual-Glo^®^ Luciferase Assay System) was used to determine the ARE-driven promoter activity. The remaining cells were divided into 4 groups: NT, niacin (1 mM), LPS (1 μg/mL), and LPS + niacin. Using the transfection reagent, pGL4.37 (luc2P/ARE/Hygro vector) and pGL4.74 (hRluc/TKvector), pNF-кB-luc and pGL4.74 (hRluc/TKvector) plasmids were transfected into cells, respectively. After the treatment of niacin (1 mM) and LPS (1 μg/mL), a dual-luciferase reporter assay system (Dual-Glo^®^ Luciferase Assay System) was put into use to determine the NRF-2/ARE and NF-кB promoter activity.

### 4.11. Determination of Adenosine Triphosphate (ATP), Adenosine Diphosphate (ADP) and Adenosine Monophosphate (AMP) Levels by High-Performance Liquid Chromatography

When the cell density reached 60%–80%, niacin, GPR109A-shRNA1, or niacin + GPR109A-shRNA1 were added and cultured for 24 h. The cells were collected into a 1.5-mL centrifuge tube, 0.4 mol/L HClO_4_ was added and then broken up by ultrasound, and centrifuged to collect the supernatant. Then, the same amount of K_2_HPO_4_ was added into the supernatant, the pH was adjusted to 6.5, and the supernatant was centrifugated and collected for detection. The adenosine triphosphate (ATP), adenosine diphosphate (ADP), and adenosine monophosphate (AMP) levels in the BMECs were determined by high-performance liquid chromatography using a Sepax Bio-C18 column (4.6 mm i.d. × 250 mm, 5 μm, 200 A) and a UV detector at a wavelength of 254 nm (bandwidth: 16 nm), as previously described [52]. The mobile phase used for separation of ATP, ADP, and AMP was NaH_2_PO_4_ (50 mmol/L, 1% methanol).

### 4.12. H&E Staining

Dairy cow mammary glands were fixed in 4% formalin and dehydrated in ethanol. After paraffin embedding, 5-μm sections were cut and stained with hematoxylin and eosin (H&E). The H&E-stained sections were examined under a light microscope [51].

### 4.13. Electron Microscopy

The cells were treated with niacin or LPS + niacin. The cells were washed twice with ice-cold PBS, fixed with 2.5% glutaraldehyde in 0.15 mM sodium cacodylate at 4 °C overnight, and then postfixed in 2% osmium tetroxide. All samples were dehydrated in ethanol and embedded in epoxy resin. Then, ultrathin sections (80 nm) of adherent cells were obtained using an ultramicrotome (EM UC7; Leica). The sections were counterstained with uranyl acetate and lead citrate and observed using a JEM SX 100 electron microscope (Jeol, Tokyo, Japan) to capture images.

### 4.14. Immunohistochemistry

The IHC detection of GPR109A was performed on paraffin sections of the mammary gland: the mammary gland tissue was perfused with 4% freshly prepared paraformaldehyde and 0.01 mol/L PBS buffer (pH 6.0) for 5 min. The mammary glands were cut into 1–2 mm frontal or sagittal slices and embedded in paraffin using an automated vacuum tissue processor (Leica ASP300, Germany). For experiments, 5-μm sections were cut and mounted on specific slides. For antigen retrieval, the protocol recently described by Pizarro et al. [53] and Strober et al. was used. Briefly, the deparaffinized and rehydrated sections were subjected to 5-min digestion with 0.01% trypsin (Sigma) in PBS and an additional microwave treatment in 10 mmol/L citrate buffer (pH 6.0) for 15 min at 720 W, followed by a 20 min cooling period in the same buffer. After pre-treatment, sections were treated with 3% hydrogen peroxidase for 15 min to block endogenous peroxidase activity and with 0.5% blocking reagent for 30 min to reduce the non-specific number of reactions. The sections were incubated for 12 h at 4 °C with anti-rat-HM74 (GPR109A) (Santa Cruz, USA; diluted 1:400) in blocking buffer. Sections were washed in PBS, and a biotinylated linking antibody solution (Biozol, Eching, Germany) was applied (100 μL/section) for 20 min at 37 °C. The sections were then washed in PBS before the application of the HRP-conjugated streptavidin (100 μL/section, 15 min) at 37 °C. Slides were stained with DAB (IBL, Hamburg, Germany) after being washed in PBS for 5 min, then washed with PBS and DAB (BIOS) for 5 min and counterstained with hematoxylin. For the negative controls, the primary antibody was replaced with the corresponding affinity-purified pre-immune IgG.

### 4.15. Immunofluorescence

BMECs were fixed with Immunol staining fix solution (P0098, Beyotime, China) at room temperature for 10 min. Cells were incubated with PBS containing 5% donkey serum albumin to block nonspecific interactions and were incubated overnight at 4 °C with primary antibodies diluted in 5% donkey serum albumin solution. Following the overnight incubation, the cells were washed three times with PBS and exposed to secondary antibodies in 5% donkey serum albumin solution for 1 h at room temperature. The controls were treated in the same manner, except that no antibodies were included in the solution. Finally, the cells were washed three times in PBS and stained with DAPI for 5 min. Images of the stained sections were obtained using a confocal laser-scanning microscope (TCS SP5; Leica, Mannheim, Germany).

### 4.16. Co-Immunoprecipitation

The culture medium was carefully removed from the confluent cells. The cells were washed once with PBS. An ice-cold IP lysis/wash buffer (250 μL) was added to the cells, which were then incubated on ice for 5 min with periodic mixing. The lysate was transferred to a microcentrifuge tube and centrifuged at ~13,000× *g* for 10 min to pelletize the cell debris. The supernatant was added to a new tube for the protein concentration determination and further analysis. The cell lysates were combined, and 5 μg of P62 was added to each sample in a microcentrifuge tube. The suggested amount of the total protein per IP reaction was 500–1000 μg, and the amounts were determined by a Pierce BCA Protein Assay (ThermoFisher, Rockford, IL, USA). The antibody/lysate solution was diluted to 500 μL with IP lysis/wash buffer and incubated for 1–2 h at room temperature or overnight at 4 °C to form the immune complex. Twenty-five microlitres (0.25 mg) of Pierce protein A/G magnetic beads were placed into a 1.5-mL microcentrifuge tube to which 175 µL of IP lysis/wash buffer was added and gently vortexed to mix. The tube with the beads was placed into a magnetic stand to enable the beads to collect against the sides of the tube. The supernatant was removed and discarded, and 1 mL of IP lysis/wash buffer was added to the tube. The tube was inverted several times or gently vortexed for 1 min to mix the contents. The beads were collected with a magnetic stand, and the supernatant was removed and discarded. The antigen sample/antibody mixture (Section B) was added to the tube containing the pre-washed magnetic beads, which were incubated at room temperature for 1 h with mixing. The beads were collected with a magnetic stand and save for analysis; the unbound sample was removed. Five hundred microlitres of IP lysis/wash buffer was added to the tube and gently mixed. The beads were collected, and the supernatant discarded. The beads were washed twice, and 500 µL of ultra-pure water was added to the tube and gently mixed. The beads were collected on a magnetic stand, and the supernatant was discarded. For the low pH elution, 100 µL of elution buffer was added to the tube, which was then incubated at RT with mixing for 10 min. The beads were magnetically separated, and the supernatant containing the target antigen was saved. To neutralize the low pH, 10 µL of neutralization buffer was added to each 100 µL of the eluate. The collected supernatant was used for the Western blot assay.

### 4.17. Western Blot Analysis

The total protein was isolated from the BMECs and bovine mammary glands using RIPA lysis buffer (Beyotime, Shanghai, China; 50 mM Tris, pH 7.4; 150 mM NaCl; 1% Triton X-100; 1% sodium deoxycholate; 0.1% SDS; sodium orthovanadate; sodium fluoride; ethylenediaminetetraacetic acid, leupeptin; and 1 mM PMSF). The lysates were centrifuged at 12,000× *g* for 5 min at 4 °C. The protein concentrations were determined using an enhanced BCA protein assay kit (Beyotime, Shanghai, China). The Western blot assay was performed in accordance with the experimental methods that our group has previously used [51].

### 4.18. Statistical Analysis

The images were generated using GraphPad Prism software (La Jolla, CA, USA). Based on extensive experience with cow models of LPS and the planned analytical framework, we estimated the number of dairy cows needed per group to detect the effects of interest at a *p* < 0.05 level. The number of technical or biological replicates (independent experiments for cell culture or individual dairy cows for in vivo experiments) in each group is specified in the figure legends. All data are presented as the mean ± SD. Two-way repeated-measures ANOVA (based on GLM) was used for comparison in the dairy cow experiment. ANOVA (general linear model) was used for comparisons of more than two groups. In cases where overall F-tests were significant (*p* < 0.05), posthoc comparisons using Tukey’s method of adjustment were conducted to determine the specific significant pairwise differences. Analyses were performed using GraphPad Prism 8.02 software.

## Figures and Tables

**Figure 1 ijms-21-03321-f001:**
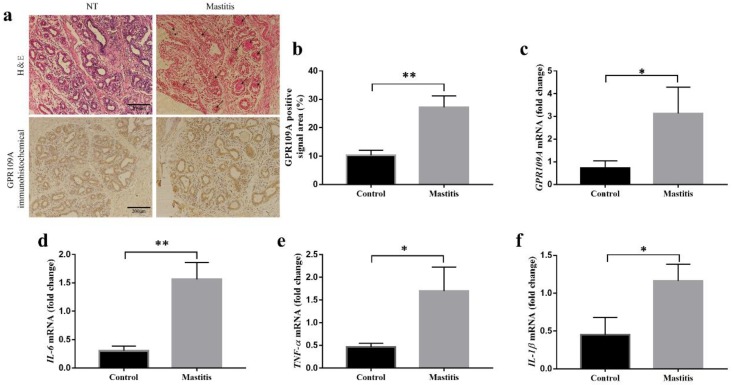
The relative expression of GPR109A, *IL-6, TNF-α*, and *IL-1β*. The mammary glands were collected from healthy dairy cows and mastitis dairy cows (*n* = 6). (**a**,**b**) The results of hematoxylin and eosin (H&E) staining and immunohistochemistry assays in the control and mastitis dairy cows. (**c**) The gene levels of *GPR109A* in the control and mastitis dairy cows. (**d**–**f**) The expression of pro-inflammatory factors in the control and mastitis dairy cows. The mRNA levels of *GPR109A* (**g**), *IL-6* (**d**), *TNF-α* (**e**), and *IL-1β* (**f**) were normalized to the level of *β-actin*. The values are presented as the means ± SD (∗ *p* < 0.05 and ∗∗ *p* < 0.01).

**Figure 2 ijms-21-03321-f002:**
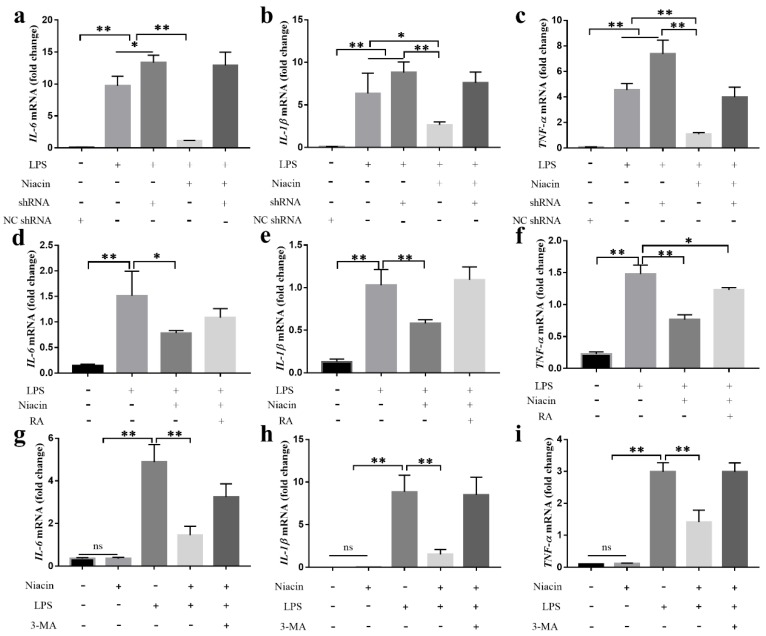
Effects of autophagy, GPR109A, and NRF2 on BMECs inflammation. (**a**–**i**) The bovine mammary epithelial cells (BMECs) were pre-treated with niacin and NC shRNA, shRNA, retinoic acid (RA), or 3-methyladenine (3-MA) for 1 h and then stimulated with LPS for 24 h. The mRNA levels of *IL-6, IL-1β*, and *TNF-α* were determined by qRT-PCR. The mRNA levels of *IL-6, IL-1β*, and *TNF-α* were normalized to the level of *β-actin*. The values are presented as the means ± SD (∗ *p* < 0.05 and ∗∗ *p* < 0.01).

**Figure 3 ijms-21-03321-f003:**
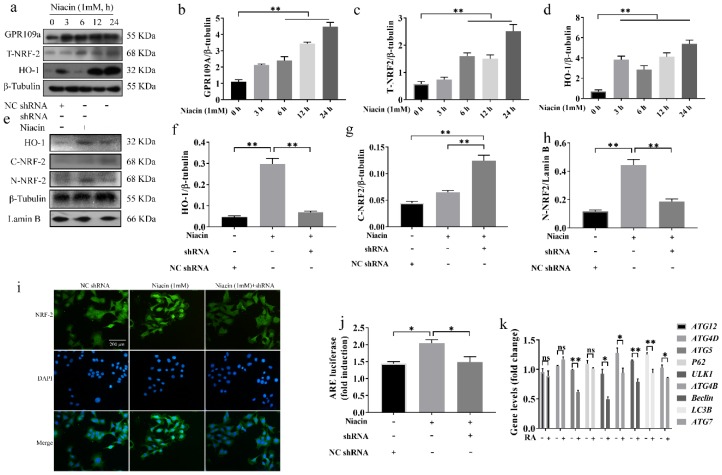
Niacin can activate the GPR109A/NRF2/autophagy signal pathway. The cells were collected at 0, 3, 6, 12, and 24 h to extract the total protein. The total protein was prepared and subjected to Western blotting using GPR109A, NRF-2, HO-1, and β-tubulin antibodies. T-NRF-2 means total NRF-2. (**a**–**d**) The protein levels of GPR109A, NRF-2, and HO-1. The cells from different experimental groups were treated with niacin or shRNA+niacin for 24 h, and then, the total protein was collected. N-NRF-2 means NRF-2 in the nucleus. C-NRF-2 means NRF-2 in the cytoplasm. (**e**–**h**) The protein levels of GPR109A, C-NRF-2, N-NRF-2, and HO-1. Each immunoreactive band was digitized and expressed as a ratio of the β-tubulin level. (**i**) The immunofluorescence results of the assay for NRF-2. The scale length in the figure is 200 μM. (**j**) The relative fluorescence intensity of NRF-2/ARE. The mRNA levels were determined by qRT-PCR. (**k**) The mRNA levels of *ATG12, ATG4D, p62, ATG5, ULK1, ATG4B, Beclin, LC3B*, and *ATG7* were normalized to the level of *β-actin*. The values are presented as the means ± SD (∗ *p* < 0.05 and ∗∗ *p* < 0.01).

**Figure 4 ijms-21-03321-f004:**
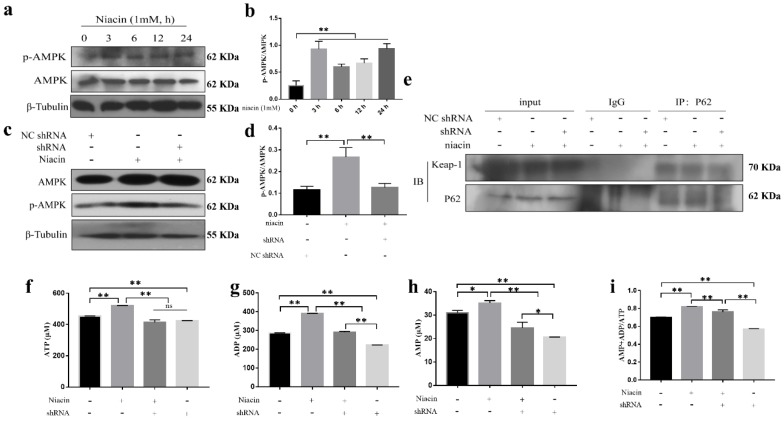
GPR109A regulates energy metabolism, AMPK phosphorylation, and P62 interact with Keap-1 in BMECs. The cells were collected at 0, 3, 6, 12, and 24 h to extract the total protein. (**a**,**b**) The protein levels of p-AMPK. The cells from different experimental groups were treated with niacin or shRNA + niacin for 24 h, and then, the total protein was collected. The cell lysates were prepared and subjected to Western blotting using AMPK and p-AMPK antibodies. (**c**,**d**) The protein levels of p-AMPK. Each immunoreactive band was digitized and expressed as a ratio of the β-tubulin. (**e**) The BMECs were pre-treated with niacin, niacin+shRNA, or shRNA for 24 h. The cells were then collected, and the ATP, ADP, and AMP levels were detected by liquid chromatography (*n* = 3). (**f**–**h**) of the ATP, ADP, and AMP content in the BMECs after adding niacin, niacin + shRNA or shRNA. (**i**) The ratio of (AMP + ADP)/ATP. The values are presented as the means ± SD (ns means no difference, ∗ *p* < 0.05 and ∗∗ *p* < 0.01).

**Figure 5 ijms-21-03321-f005:**
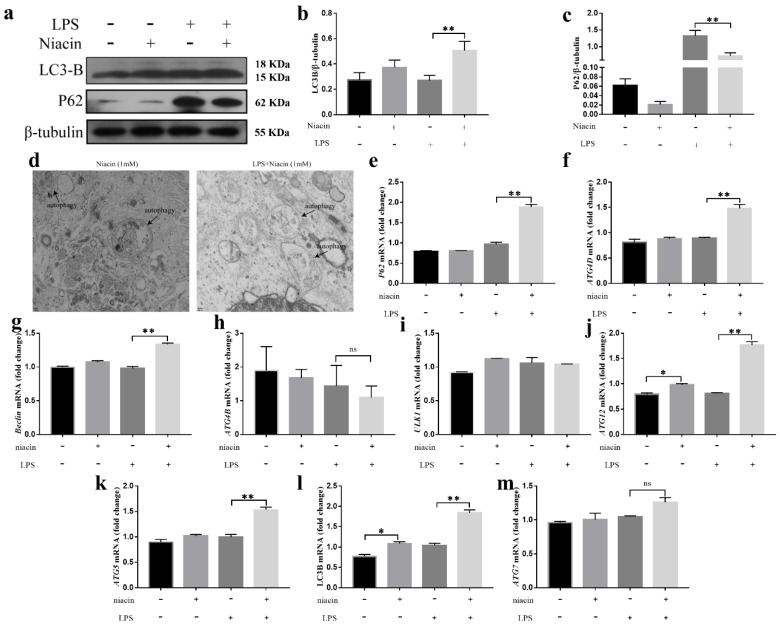
GPR109A alleviates inflammation of BMECs by regulating autophagy. Cells from different experimental groups were treated with niacin, LPS, or niacin+LPS for 24 h, and then, the total protein was collected. The cell lysates were prepared and subjected to Western blotting using β-tubulin (**a**), LC3B (**a**,**b**), and P62 (**a**,**c**) antibodies. Each immunoreactive band was digitized and expressed as a ratio of the β-tubulin. (**d**) The left panel shows 6000 × 120 V; the right panel shows is 12,000 × 120 V. The results revealed by the electron microscopy showed that niacin could induce autophagy in the non-treated BMECs and LPS-treated BMECs. (**e**–**m**) The mRNA levels were determined by qRT-PCR. The mRNA levels of *ATG12, ATG4D*, *p62*, *ATG5, ULK1, ATG4B, Beclin, LC3B*, and *ATG7* were normalized to the level of *β-actin*. Values are presented as the means ± SD (∗ *p* < 0.05 and ∗∗ *p* < 0.001).

**Figure 6 ijms-21-03321-f006:**
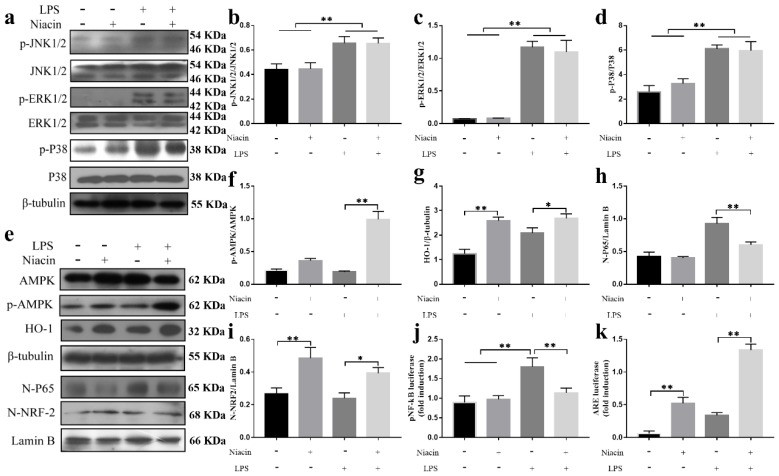
Effect of GPR109A on the AMPK/NRF-2/HO-1, P38, JNK1/2, and ERK1/2 signaling pathways. The cells from different experimental groups were treated with niacin, niacin, or LPS for 24 h, and then, the total protein was collected. N-P65 indicates P65 in the nucleus. N-NRF-2 means NRF-2 in the nucleus. The cell lysates were prepared and subjected to Western blotting using antibodies for P38 (**a**,**d**), p-P38 (**a**,**d**), JNK1/2 (**a**,**b**), p-JNK1/2 (**a**,**b**), ERK1/2 (**a**,**c**), p-ERK1/2 (**a**,**c**), AMPK (**a**,**f**), p-AMPK (**a**,**f**), HO-1 (**a**,**g**), β-tubulin (**e**), N-P65 (**a**,**h**), N-NRF-2 (**a**,**i**–**k**), and laminin B (**e**).

**Table 1 ijms-21-03321-t001:** The primer sequenceof the genes.

Gene		Primer Sequence
*GPR109A*	Forward	5′-ACCTCGTTCCCCGAACCTTG-3′
	Reverse	5′-CGGCAGCACTTTGGCAATGA-3′
*TNF-α*	Forward	5′-ACGGGCTTTACCTCATCTACTC-3′
	Reverse	5′-GCTCTTGATGGCAGACAGG-3′
*IL-6*	Forward	5′-ATGCTTCCAATCTGGGTTC-3′
	Reverse	5′-TGAGGATAATCTTTGCGTTC-3′
*IL-1β*	Forward	5′-AGGTGGTGTCGGTCATCGT-3′
	Reverse	5′-GCTCTCTGTCCTGGAGTTTGC-3′
*β-actin*	Forward	5′-TCACCAACTGGGACGACA-3′
	Reverse	5′-GCATACAGGGACAGCACA-3′
*ATG4B*	Forward	5′-AGGTGGACGCAGCGGAAGAG-3′
	Reverse	5′GACAGCCAGCTTCTTGAGGACTTG-3′
*Beclin*	Forward	5′-TGCACAGACACTCTCCTAGACCAG-3′
	Reverse	5′-ATCAGCCTCTCCTCCTCTAATGCC-3′
*ATG12*	Forward	5′-AAGATGGCTGAGGAGCAGGAGTC-3′
	Reverse	5′-GGAGACCTCGGTAGGCACTTCAG-3′
*ATG7*	Forward	5′-AAGGTTGTGTCTGCCAAGTGTCTG-3′
	Reverse	5′-TTGTCCACGAACGTGATGTGTCTG-3′
*LC3B*	Forward	5′-GCAGGCCACCGTTCACTCTTG-3′
	Reverse	5′-ATGCAGCAGGAAGAGCAGATTGG-3′
*ATG4D*	Forward	5′-GGCTGAATGGAAGTCCGTGGTC-3′
	Reverse	5′-GTAGCCGATGAAGTACAGCGAGTG-3′
*ATG5*	Forward	5′-AGCATCATCCCGCAACCAAC-3′
	Reverse	5′-GACCAGCCCTAGTGCCCTTA-3′
*P62*	Forward	5′-GTGATCTGTGACGGCTGTAACGG-3′
	Reverse	5′-AGGCGGAGCATAGGTCGTAGTC-3′
*ULK1*	Forward	5′-GCATCGGCACCATCGTGTACC-3′
	Reverse	5′-GGACCAGCGTCTTGTTCTTCTCG-3′

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
