# Peer review of "Niacin Alleviates Dairy Cow Mastitis by Regulating the GPR109A/AMPK/NRF2 Signaling Pathway"

_ijms, 2020, doi:10.3390/ijms21093321_

Round 1

Reviewer 1 Report

Suggestions to the authors:

  • In the “introduction” and in the “materials and methods” section there are bibliographic references that are not properly numbered
  • The literature mentioned in the introduction must be associated by a valid bibliographic reference
  • In the “materials and methods” section, the primer sequences used must be listed in a table in order to simplify reading
  • In the “materials and methods” section, the dilutions of primary and secondary antibodies used in western blot analyses must be added
  • The rationale about niacin dose and treatment period should be added
  • More information about the impact of niacin treatment on milk production and quality, on lipid metabolism and oxidative stress should be added in the discussion section.

Author Response

Suggestions to the authors:

Thank you very much for your valuable comments. I have finished the revision of the paper carefully. Hope to get your approval. Here is my proof of language modification.

  1. In the “introduction” and in the “materials and methods” section there are bibliographic references that are not properly numbered

A:Thank you very much for your comments. I have revised it in the paper (introduction: line 54, 56, 58, 64, 65) (materials and methods: line 98, 122, 127, 136, 160, 164, 235).

  1. The literature mentioned in the introduction must be associated by a valid bibliographic reference

A:Thank you very much for your comments. I have revised it in the paper.

  1. In the “materials and methods” section, the primer sequences used must be listed in a table in order to simplify reading

A:Thank you very much for your comments. I have revised it in the paper (line 138).

  1. In the “materials and methods” section, the dilutions of primary and secondary antibodies used in western blot analyses must be added

A:Thank you very much for your comments. I have revised it in the paper (line 75-86).

  1. The rationale about niacin dose and treatment period should be added

A:Thank you very much for your comments. I have explained in the materials and methods section why this dose and treatment cycle was used (96-98).

At present, the treatment cycle of mastitis in dairy farm is generally 7-10 days, so 7 days is selected as the treatment cycle in this study, and the dosage is selected according to the previous research and the results of my pre-experiment.

  1. More information about the impact of niacin treatment on milk production and quality, on lipid metabolism and oxidative stress should be added in the discussion section.

A:Thank you very much for your comments. I've added to the discussion about oxidative stress and milk production (line 485-491).

Reviewer 2 Report

This manuscript reports a study entitled “ Niacin alleviates dairy cow mastitis by regulating GPR109A/AMPK/NRF2 signaling pathway and autophagy” is an interesting topics and great concerns for dairy industry. Authors hypothesized to show mechanism of niacin action on alleviating mastitis on mammary epithelial cells. Authors used various biochemical, molecular and cellular microscopy techniques to show the evidence of multiple signaling pathways starting from inflammation to autophagy by investigating multiple regulatory proteins involved in the said pathways.
The protection of bovine mammary epithelial cells (BMECs) from mastitis and its health is very important. Therefore this research discovery would have profound implications and impact on dairy industry. However, this reviewer believes that this study lacks the robustness to justify the extraordinary claims. To a minimal, the authors should provide convincing data on isolation procedure of BMECs to experimental protein chemistry, immunohistochemistry and microscopy data through major revision before accepting the manuscript for publication.

Specific Major Comments:
Page 3, first line – “Serum and milk of each group were collected”. I guess authors’ means to collect blood from the cow not the serum. What is the procedure of collection of blood such as venipuncture etc. and from where? Please explain in materials and methods sections.

Page 3- Plasmid construction- Two GPR109A-shRNA used

  1. GPR109A-shRNA1
  2. GPR109A-shRNA11

Which one is correct one (a) or (b)?

I guess typographical error. Authors need to correct it in revised version. Also authors need to provide catalogue number that may be required by investigators in future study.

Page 3, ELISA- Authors should provide the ELISA kits catalogue number, that will provide information for readers any investigators to use on their specific experiments.

Page 3, Cell culture- How bovine mammary cells (BMECs) were isolated and there is no references provided. This is very important to address because most of the experiments are based on the BMECs. Authors need to provide explanation on isolation procedure of BMECs and any available references. Furthermore, authors need to show the BMECs specific markers to show the homogeneity of BMECs.

Page 4, Determination of ATP………liquid chromatography- Authors need to explain briefly how samples were processed for HPLC and what references used.

Page 4, H & E staining- what is the source of dairy cow mammary glands tissue? Is authors’ sacrifice the study animal to obtain mammary glands?

Page 5, immunofluorescence – What is the dilution concentration of primary and secondary antibody? What is the catalogue number?

Page 5, Co-immunoprecipitation- Line 6 of this co-immunoprecipitation section “ and 2-10 mg of p62”. 2-10 mg of p62 used seems a very wide range, what specific amount used for co-IP? Be specific.

Page 5, Western Blot Analysis- What are the primary and secondary antibody concentrations used? Authors cited previous working references [21]. But new antibody working concentration should be mentioned.

Page 6, Expression of GPR109A………….and mastitis dairy cows- This is an excellent findings described in figure 1 a-f. However, authors failed to mention where and how they obtained mammary gland of cows to perform this study. Figure 1a (upper) authors claimed about neutrophil infiltration in mastitis cow compared to NT (need to mention what is NT) cow. I guess non-treated. In science no none should guess. It is an experimental study. Authors need to describe clearly each terminology they have used in the articles.

But my concern is neutrophil exists in six different stages myoblast to matured segmented neutrophil. What specific stages of these neutrophils are to determine the disease conditions? A best method is to use blood smear.

Figure 1a (lower panel) what concentration of GPR109A antibody used and need brief description of experimental procedure for reproducibility. This reviewer strongly suggests performing GPR109A expression in mammary gland tissue through conventional western blot techniques to support the findings.

Page 8, Activation of GPR109A………….signaling pathways in BMECs- The results in Fig. 3a GPR109A showed overexpression induced by niacin. The expression of housekeeping protein b-tubulin is not well convincing. It is a general notion that housekeeping protein expression should have clear band intensity. This reviewer suggests showing better quality b-tubulin bands in western blot. (Authors may consider the data presented in the reference given here in Fig3 ref. Sun J, et. al. Int J Mol Sci. 2017; 18(12): 2621. Published 2017 Dec 5. Doi: 10.3390/ijms18122621) from same group.

Figure 3e. The NRF2 expression in nuclear and cytosolic fraction well characterized. However, authors fail to describe how they separate nuclear and cytosolic fractions. It is neither described in materials and methods nor cited any references. (Couple of reference given here authors attention Antalis, T.M et. al Nucleic Acids Res. 199119, 4301, and Giri et. al Int. J. Mol. Sci. 201920(18), 4559).

Figure 3i. The immunofluorescence pictures resolutions are poorly presented. It should be showed in higher magnification with clear vision of cell structure. This reviewer does not recommend this figure for publications.

Figure 5 e-m. Page number 10. The authors showed the transcripts level of ATG4D, Beclin, ATG12, ATG5 and LC3B (Fig 5e, f, g, j, k, l). The level of ULK1 is not sufficient to claim autophagy. Authors need to show and protein level as well.

Specific Minor Comments:

Authors should revise, as there are missing spaces, abbreviations (NT), superscripts, and italics. Few of missing spaces are given below.

Introduction section-

Page 1, Line 5, spaces in periods “$35 billion annually .”

Page 2, Line 31, spaces in periods “glycometabolism and inflammation . ”

Page 2, Line 32, spaces in periods “ inhibiting cell survival .”

Materials and Methods section

Page 3, Line 15, spaces in words “ofacells .”

Page 3, Line 15, correct this “was 0.3 × 105.”

Page 6, Line 32, convert it to italics “ in vivo .”

Results Sections-

Page 8, Line 19, spaces in periods “autophagy-related genes .”

Page 9, Line 18- 19, Need references “Previous studies have shown that the ratio of (AMP+ADP)/ATP is the key signal for activating AMPK .”

Discussion sections-

Page 12, Line 20, spaces in periods “mediators in adipocytes or monocytes . ”

Page 13, Line 5, spaces in periods “renewal of some organelles . ”

Page 13, Line 8, spaces in periods “injury in mice . ”

Page 13, Line 14, spaces in periods “promotes NRF-2 nuclear import . ”

Once these issues are addressed, then I would happy to recommend this manuscript for publications.

Author Response

Reviewer2

This manuscript reports a study entitled “ Niacin alleviates dairy cow mastitis by regulating GPR109A/AMPK/NRF2 signaling pathway and autophagy” is an interesting topics and great concerns for dairy industry. Authors hypothesized to show mechanism of niacin action on alleviating mastitis on mammary epithelial cells. Authors used various biochemical, molecular and cellular microscopy techniques to show the evidence of multiple signaling pathways starting from inflammation to autophagy by investigating multiple regulatory proteins involved in the said pathways.
The protection of bovine mammary epithelial cells (BMECs) from mastitis and its health is very important. Therefore this research discovery would have profound implications and impact on dairy industry. However, this reviewer believes that this study lacks the robustness to justify the extraordinary claims. To a minimal, the authors should provide convincing data on isolation procedure of BMECs to experimental protein chemistry, immunohistochemistry and microscopy data through major revision before accepting the manuscript for publication.

A: Thank you very much for your valuable comments. And we have made serious improvements to the article. Your comments are of great help to us.

Specific Major Comments:
1. Page 3, first line – “Serum and milk of each group were collected”. I guess authors’ means to collect blood from the cow not the serum. What is the procedure of collection of blood such as venipuncture etc. and from where? Please explain in materials and methods sections.

A: Thank you very much for your comments. We collected the peripheral blood from the tail vein of the dairy cow with a 5 mL syringe and centrifuged it at 3000 rpm for 25 min to separate the serum. We also collected milk through the mammary gland of dairy cow, each of which collected about 10 mL, centrifuged at 3000 rpm for 25 min, and separated whey (line 100-103).

We collected the peripheral blood from the tail vein of the dairy cow with a 5 mL syringe and centrifuged it at 3000 rpm for 25 min to separate the serum. We also collected milk through the mammary gland of dairy cow, each of which collected about 10 mL, centrifuged at 3000 rpm for 25 min, and separated whey.

  1. Page 3- Plasmid construction- Two GPR109A-shRNA used
  1. GPR109A-shRNA1
  2. GPR109A-shRNA11

Which one is correct one (a) or (b)?

I guess typographical error. Authors need to correct it in revised version. Also authors need to provide catalogue number that may be required by investigators in future study.

A: Your question is very important. I'm sorry that this part has been corrected in the paper. We used GPR109A-shRNA1 plasmid (line 107, 108).

  1. Page 3, ELISA- Authors should provide the ELISA kits catalogue number, that will provide information for readers any investigators to use on their specific experiments.

A: Thank you very much for your comments. I have added the catalogue number in the paper (line 110).

The protein levels of TNF-α (Cat. H-80095), IL-6 (Cat. H-80071) and IL-1β (Cat. H-78807) in serum and milk were determined by ELISA kits according to the manufacturer’s instructions (Hengyuan, Shanghai, China).

  1. Page 3, Cell culture- How bovine mammary cells (BMECs) were isolated and there is no references provided. This is very important to address because most of the experiments are based on the BMECs. Authors need to provide explanation on isolation procedure of BMECs and any available references. Furthermore, authors need to show the BMECs specific markers to show the homogeneity of BMECs.

A: Your comments are very critical. I have described the experimental method in the article and added the reference. We also stained CK-18 in BMECs (Fig 1) (line 115-122).

The tissue extracted from the mammary gland of Holstein cows in the middle of lactation was cut into 1 mm small pieces and incubated in the humidified air containing 5% CO2 at 37 ℃. When cells migrate from the tissue covering 80% of the bottom, the tissue is removed. Epithelial cells and fibroblasts were isolated with 0.25% trypsin and 0.15% trypsin plus 0.02% EDTA. The dispersed cells were inoculated on cell culture bottle (Corning) in DMEM (GIBCO). In the basic medium, 5 mg/mL transferrin, 5 mg/mL insulin, 5 mg/mL prolactin, 1 mg/mL hydrocortisone (sigma), 10 ng/mL epidermal growth factor, 1% glutamine, 1% penicillin, 1% streptomycin and 10% fetal bovine serum (Clark) were added.

Fig 1. CK-18 in BMECs. Immunofluorescence of CK-18 in BMECs.

  1. Page 4, Determination of ATP………liquid chromatography- Authors need to explain briefly how samples were processed for HPLC and what references used.

A: Thank you very much for your comments. I have added the experimental steps and references to the paper (line 154-157).

Collect the cells into a 1.5mL centrifuge tube, add 0.4 mol/L HClO4, and then break them up by ultrasound, and centrifuged to collect supernatant. Add the same amount of K2HPO4 into the supernatant, adjust the pH to 6.5, centrifugate and collect the supernatant for detection.

  1. Page 4, H & E staining- what is the source of dairy cow mammary glands tissue? Is authors’ sacrifice the study animal to obtain mammary glands?

A:Thank you very much for your question. All the mammary gland of dairy cows come from the healthy and mastitis cows in the middle stage of lactation. We use the method of puncture to collect the mammary gland tissue of dairy cow. After puncture, we need to carry out the treatment of anti-bacterial and anti-inflammatory, so as to facilitate the rapid recovery of dairy cow.

  1. Page 5, immunofluorescence – What is the dilution concentration of primary and secondary antibody? What is the catalogue number?

A:Thank you very much for your comments. I have added dilution concentration and catalogue number to the paper (line 80-81).

  1. Page 5, Co-immunoprecipitation- Line 6 of this co-immunoprecipitation section “ and 2-10 mg of p62”. 2-10 mg of p62 used seems a very wide range, what specific amount used for co-IP? Be specific.

A:Thank you very much for your comments. I have added the specific amount in the paper (line 207).

  1. Page 5, Western Blot Analysis- What are the primary and secondary antibody concentrations used? Authors cited previous working references [21]. But new antibody working concentration should be mentioned.

A:Thank you very much for your comments. I have added dilution concentration (line 75-86).

  1. Page 6, Expression of GPR109A………….and mastitis dairy cows- This is an excellent findings described in figure 1 a-f. However, authors failed to mention where and how they obtained mammary gland of cows to perform this study. Figure 1a (upper) authors claimed about neutrophil infiltration in mastitis cow compared to NT (need to mention what is NT) cow. I guess non-treated. In science no none should guess. It is an experimental study. Authors need to describe clearly each terminology they have used in the articles.

A: Thank you very much for your comments. I have revised these two points in the paper (line 251-252, 254-256, 261-262).

A small tissue of mammary gland of healthy cow and mastitis cow was taken out by puncture.

The H&E staining analysis revealed that a large number of neutrophils had infiltrated the mammary acini of mastitis dairy cows, but neutrophils did not invade the mammary acini of healthy cows (Control group) (Fig 1a).

Second, the protein and mRNA levels of GPR109A were detected by immunohistochemistry and qRT-PCR for three times, respectively.

But my concern is neutrophil exists in six different stages myoblast to matured segmented neutrophil. What specific stages of these neutrophils are to determine the disease conditions? A best method is to use blood smear.

A: Thank you very much for your comments. We used California Mastitis Test (CMT) to screen dairy cows in early stage, The CMT divides the mastitis into five stages according to the state of milk in dairy cows. We only chose positive dairy cows. So we didn't focus on the stage of neutrophils, just to determine whether cows are in mastitis. The following table shows the classification of CMT.

Table 1. The judging method and standard of CMT

Reaction determination

Milk tested

Reaction state

Somatic cell counts (10000 / ml)

-

Negative

The milk mixture is liquid. When the inspection disk is tilted, the flow is smooth and there is no clot.

0-20

±

Suspicious

The milk mixture is liquid, and there is a small amount of sediment at the bottom of the plate, which disappears when shaking.

20-50

+

Weak positive

There is a small amount of sticky sediment on the bottom of the pan, which does not form a gel like form. When it is shaking, the sediment is scattered on the bottom of the dish and has certain viscosity.

++

Positive

All of them are gelatinous and have a certain viscosity. They are concentrated in the center when rotating, and are not easy to disperse.

80-500

+++

Strong positive

Most or all of the mixture forms obvious gelatinous precipitate, which is thick and almost completely adheres to the bottom of the plate. When rotating and shaking, the precipitate gathers in the center and is difficult to disperse.

>500

Figure 1a (lower panel) what concentration of GPR109A antibody used and need brief description of experimental procedure for reproducibility. This reviewer strongly suggests performing GPR109A expression in mammary gland tissue through conventional western blot techniques to support the findings.

A: Thank you very much for your comments. The concentration of GPR109A in immunohistochemistry was 1:50 (line 82). I've added this description to the material method. I've also added brief description of experimental procedure for reproducibility in the result part. We also used WB technology to detect the protein level of GPR109a in two groups of mammary gland. We can find that the protein levels of GPR109A in mastitis group is significantly higher than that of Control group (Fig 2).

Fig 2. Protein level of GPR109A in mammary gland. (a, b) Protein levels of GPR109A. The values are presented as the means ± SD (∗p<0.05 and ∗∗p<0.01).

  1. Page 8, Activation of GPR109A………….signaling pathways in BMECs- The results in Fig. 3a GPR109A showed overexpression induced by niacin. The expression of housekeeping protein b-tubulin is not well convincing. It is a general notion that housekeeping protein expression should have clear band intensity. This reviewer suggests showing better quality b-tubulin bands in western blot. (Authors may consider the data presented in the reference given here in Fig3 ref. Sun J, et. al. Int J Mol Sci. 2017; 18(12): 2621. Published 2017 Dec 5. Doi: 10.3390/ijms18122621) from same group.

A: Thank you very much for your comments. I've replaced a clearer β-tubulin bands in the paper (Fig 3).

Fig 3 Niacin can activate GPR109A/NRF2/autophagy signal pathway. Niacin can activate GPR109A/NRF-2/autophagy signal pathway. The cells were collected at 0 h, 3 h, 6 h, 12 h and 24 h to extract the total protein. The total protein was prepared and subjected to western blotting using GPR109A, NRF-2, HO-1 and β-tubulin antibodies. T-NRF-2 means total NRF-2. (a-d) The protein levels of GPR109A, NRF-2 and HO-1. The cells from different experimental groups were treated with niacin or shRNA+niacin for 24 h, and then, the total protein was collected. N-NRF-2 means NRF-2 in the nucleus. C-NRF-2 means NRF-2 in the cytoplasm. (e-h) The protein levels of GPR109A, C-NRF-2, N-NRF-2 and HO-1. Each immunoreactive band was digitized and expressed as a ratio of the β-tubulin level. (i) The immunofluorescence results of the assay for NRF-2. (j) The relative fluorescence intensity of NRF-2/ARE. The mRNA levels were determined by qRT-PCR. (k) The mRNA levels of ATG12, ATG4D, p62, ATG5, ULK1, ATG4B, Beclin, LC3B and ATG7 were normalized to the level of β-actin. The values are presented as the means ± SD (∗p<0.05 and ∗∗p<0.01).

Figure 3e. The NRF2 expression in nuclear and cytosolic fraction well characterized. However, authors fail to describe how they separate nuclear and cytosolic fractions. It is neither described in materials and methods nor cited any references. (Couple of reference given here authors attention Antalis, T.M et. al Nucleic Acids Res. 199119, 4301, and Giri et. al Int. J. Mol. Sci. 201920(18), 4559).

A: Thank you very much for your comments. I have added the experimental method of cytoplasm and nucleus separation to the material method in the paper (line 123-127).

Separation of nucleus and cytoplasm

We inoculated the BMECs into the cell culture dish. When the cells reached about 80%, GPR109A-shRNA, niacin or LPS were used to treat the BMECs. Cells were washed twice with cold PBS after the above treatment and subjected to nuclear and cytosolic fractionations with modification described by Antalis et al and Giri et. al.

Figure 3i. The immunofluorescence pictures resolutions are poorly presented. It should be showed in higher magnification with clear vision of cell structure. This reviewer does not recommend this figure for publications.

A: Thank you very much for your comments. I re-import Figure 3i. Now when you zoom in, you can see the cell structure clearly (Fig 4).

Fig 4. Immunofluorescence image of Nrf2 in nucleus. Immunofluorescence results of Nrf2.

Figure 5 e-m. Page number 10. The authors showed the transcripts level of ATG4D, Beclin, ATG12, ATG5 and LC3B (Fig 5e, f, g, j, k, l). The level of ULK1 is not sufficient to claim autophagy. Authors need to show and protein level as well.

 A: The questions you ask are crucial. Because Nrf2 is a transcription factor, it can combine with DNA in the nucleus to promote the transcription of downstream target genes. In our study, niacin can significantly promote Nrf2 nuclear import, so we want to know whether Nrf2 can also promote the expression of downstream autophagy related genes in the nucleus. So, we do qRT-PCR mainly to clarify that Nrf2 can regulate the expression of autophagy related genes at the transcription level. However, the specific pathway of niacin activating downstream autophagy has not been studied. But in this study, we can clearly see the formation of autophagy lysosomes in the electron microscopy results. Western blotting of LC3B and P62 also showed that niacin can activate autophagy. Therefore, our next plan is to elucidate the pathway of niacin activating autophagy. And we will continue to consider publishing our results in IJMS. Thank you again for your valuable comments.

Specific Minor Comments:

Authors should revise, as there are missing spaces, abbreviations (NT), superscripts, and italics. Few of missing spaces are given below.

Introduction section-

Page 1, Line 5, spaces in periods “$35 billion annually .”

A: Thank you very much for your comments. I have revised in the paper.

Page 2, Line 31, spaces in periods “glycometabolism and inflammation . ”

A: Thank you very much for your comments. I have revised in the paper.

Page 2, Line 32, spaces in periods “ inhibiting cell survival .”

A: Thank you very much for your comments. I have revised in the paper.

Materials and Methods section

Page 3, Line 15, spaces in words “ofacells .”

A: Thank you very much for your comments. I have revised in the paper.

Page 3, Line 15, correct this “was 0.3 × 105.”

A: Thank you very much for your comments. I have revised in the paper.

Page 6, Line 32, convert it to italics “ in vivo .”

A: Thank you very much for your comments. I have revised in the paper.

Results Sections-

Page 8, Line 19, spaces in periods “autophagy-related genes .”

A: Thank you very much for your comments. I have revised in the paper.

Page 9, Line 18- 19, Need references “Previous studies have shown that the ratio of (AMP+ADP)/ATP is the key signal for activating AMPK .”

A: Thank you very much for your comments. I have revised in the paper.

Discussion sections-

Page 12, Line 20, spaces in periods “mediators in adipocytes or monocytes . ”

A: Thank you very much for your comments. I have revised in the paper.

Page 13, Line 5, spaces in periods “renewal of some organelles . ”

A: Thank you very much for your comments. I have revised in the paper.

Page 13, Line 8, spaces in periods “injury in mice . ”

A: Thank you very much for your comments. I have revised in the paper.

Page 13, Line 14, spaces in periods “promotes NRF-2 nuclear import . ”

A: Thank you very much for your comments. I have revised in the paper.

Once these issues are addressed, then I would happy to recommend this manuscript for publications.

Round 2

Reviewer 2 Report

Reviewer2

This manuscript reports a study entitled “ Niacin alleviates dairy cow mastitis by regulating GPR109A/AMPK/NRF2 signaling pathway and autophagy” is an interesting topics and great concerns for dairy industry. Authors hypothesized to show mechanism of niacin action on alleviating mastitis on mammary epithelial cells. Authors used various biochemical, molecular and cellular microscopy techniques to show the evidence of multiple signaling pathways starting from inflammation to autophagy by investigating multiple regulatory proteins involved in the said pathways.

The protection of bovine mammary epithelial cells (BMECs) from mastitis and its health is very important. Therefore this research discovery would have profound implications and impact on dairy industry. However, this reviewer believes that this study lacks the robustness to justify the extraordinary claims. To a minimal, the authors should provide convincing data on isolation procedure of BMECs to experimental protein chemistry, immunohistochemistry and microscopy data through major revision before accepting the manuscript for publication.

A: Thank you very much for your valuable comments. And we have made serious improvements to the article. Your comments are of great help to us.

Specific Major Comments:
1. Page 3, first line – “Serum and milk of each group were collected”. I guess authors’ means to collect blood from the cow not the serum. What is the procedure of collection of blood such as venipuncture etc. and from where? Please explain in materials and methods sections.

A: Thank you very much for your comments. We collected the peripheral blood from the tail vein of the dairy cow with a 5 mL syringe and centrifuged it at 3000 rpm for 25 min to separate the serum. We also collected milk through the mammary gland of

dairy cow, each of which collected about 10 mL, centrifuged at 3000 rpm for 25 min, and separated whey (line 100-103).

We collected the peripheral blood from the tail vein of the dairy cow with a 5 mL syringe and centrifuged it at 3000 rpm for 25 min to separate the serum. We also collected milk through the mammary gland of dairy cow, each of which collected about 10 mL, centrifuged at 3000 rpm for 25 min, and separated whey.

  1. Page 3- Plasmid construction- Two GPR109A-shRNA used
  2. GPR109A-shRNA1
  3. GPR109A-shRNA11

Which one is correct one (a) or (b)?

I guess typographical error. Authors need to correct it in revised version. Also authors need to provide catalogue number that may be required by investigators in future study.

A: Your question is very important. I'm sorry that this part has been corrected in the paper. We used GPR109A-shRNA1 plasmid (line 107, 108).

Where is the catalogue number?

  1. Page 3, ELISA- Authors should provide the ELISA kits catalogue number, that will provide information for readers any investigators to use on their specific experiments.

A: Thank you very much for your comments. I have added the catalogue number in the paper (line 110).

The protein levels of TNF-α (Cat. H-80095), IL-6 (Cat. H-80071) and IL-1β (Cat. H-78807) in serum and milk were determined by ELISA kits according to the manufacturer’s instructions (Hengyuan, Shanghai, China).

Hengyuan company name is not available. Do you have contact information of this company?

  1. Page 3, Cell culture- How bovine mammary cells (BMECs) were isolated and there is no references provided. This is very important to address because most of the experiments are based on the BMECs. Authors need to provide explanation on isolation procedure of BMECs and any available references. Furthermore, authors need to show the BMECs specific markers to show the homogeneity of BMECs.

A: Your comments are very critical. I have described the experimental method in the article and added the reference. We also stained CK-18 in BMECs (Fig 1) (line 115- 122).

The tissue extracted from the mammary gland of Holstein cows in the middle of lactation was cut into 1 mm small pieces and incubated in the humidified air containing 5% CO2 at 37 °C. When cells migrate from the tissue covering 80% of the bottom, the tissue is removed. Epithelial cells and fibroblasts were isolated with 0.25% trypsin and 0.15% trypsin plus 0.02% EDTA. The dispersed cells were inoculated on cell culture bottle (Corning) in DMEM (GIBCO). In the basic medium, 5 mg/mL transferrin, 5 mg/mL insulin, 5 mg/mL prolactin, 1 mg/mL hydrocortisone (sigma), 10 ng/mL epidermal growth factor, 1% glutamine, 1% penicillin, 1% streptomycin and 10% fetal bovine serum (Clark) were added.

Fig 1. CK-18 in BMECs. Immunofluorescence of CK-18 in BMECs.

  1. Page 4, Determination of ATP.........liquid chromatography- Authors need to explain briefly how samples were processed for HPLC and what references used.

A: Thank you very much for your comments. I have added the experimental steps and references to the paper (line 154-157).

Collect the cells into a 1.5mL centrifuge tube, add 0.4 mol/L HClO4, and then break them up by ultrasound, and centrifuged to collect supernatant. Add the same amount of K2HPO4 into the supernatant, adjust the pH to 6.5, centrifugate and collect the supernatant for detection.

What is the mobile phase used for separation of ATP, ADP, and AMP?

  1. Page 4, H & E staining- what is the source of dairy cow mammary glands tissue? Is authors’ sacrifice the study animal to obtain mammary glands?

A:Thank you very much for your question. All the mammary gland of dairy cows come from the healthy and mastitis cows in the middle stage of lactation. We use the method of puncture to collect the mammary gland tissue of dairy cow. After puncture, we need to carry out the treatment of anti-bacterial and anti-inflammatory, so as to facilitate the rapid recovery of dairy cow.

  1. Page 5, immunofluorescence – What is the dilution concentration of primary and secondary antibody? What is the catalogue number?

A: Thank you very much for your comments. I have added dilution concentration and catalogue number to the paper (line 80-81).

  1. Page 5, Co-immunoprecipitation- Line 6 of this co-immunoprecipitation section “ and 2-10 mg of p62”. 2-10 mg of p62 used seems a very wide range, what specific amount used for co-IP? Be specific.

A: Thank you very much for your comments. I have added the specific amount in the paper (line 207).

  1. Page 5, Western Blot Analysis- What are the primary and secondary antibody concentrations used? Authors cited previous working references [21]. But new antibody working concentration should be mentioned.

A:Thank you very much for your comments. I have added dilution concentration (line 75-86).

  1. Page 6, Expression of GPR109A.............and mastitis dairy cows- This is an excellent findings described in figure 1 a-f. However, authors failed to mention where and how they obtained mammary gland of cows to perform this study. Figure 1a (upper) authors claimed about neutrophil infiltration in mastitis cow compared to NT (need to mention what is NT) cow. I guess non-treated. In science no one should guess. It is an experimental study. Authors need to describe clearly each terminology they have used in the articles.

A: Thank you very much for your comments. I have revised these two points in the paper (line 251-252, 254-256, 261-262).

A small tissue of mammary gland of healthy cow and mastitis cow was taken out by puncture.

The H&E staining analysis revealed that a large number of neutrophils had infiltrated the mammary acini of mastitis dairy cows, but neutrophils did not invade the mammary acini of healthy cows (Control group) (Fig 1a).

Second, the protein and mRNA levels of GPR109A were detected by immunohistochemistry and qRT-PCR for three times, respectively.

But my concern is neutrophil exists in six different stages myoblast to matured segmented neutrophil. What specific stages of these neutrophils are to determine the disease conditions? A best method is to use blood smear.

A: Thank you very much for your comments. We used California Mastitis Test (CMT) to screen dairy cows in early stage, The CMT divides the mastitis into five stages according to the state of milk in dairy cows. We only chose positive dairy cows. So we didn't focus on the stage of neutrophils, just to determine whether cows are in mastitis. The following table shows the classification of CMT.

Table 1. The judging method and standard of CMT

Reaction determination

-

Milk tested

Negative

Reaction state Somatic cell counts (10000

/ ml)

The milk mixture is 0-20 liquid. When the

± Suspicious

inspection disk is tilted, the flow is smooth and there is no clot.

The milk mixture is liquid, and there is a small amount of sediment at the bottom of the plate, which disappears when shaking.

There is a small amount of sticky sediment on the bottom of the pan, which does not form a gel like form. When it is shaking, the sediment is scattered on the bottom of the dish and has certain viscosity.

All of them are gelatinous and have a certain viscosity. They are concentrated in the

20-50

+ Weak positive

++ Positive

80-500

+++ Strong positive

center when rotating, and are not easy to disperse.

Most or all of the >500 mixture forms obvious gelatinous precipitate,
which is thick and

almost completely adheres to the bottom of the plate. When rotating and shaking, the precipitate gathers in the center and is difficult to disperse.

Figure 1a (lower panel) what concentration of GPR109A antibody used and need brief description of experimental procedure for reproducibility. This reviewer strongly suggests performing GPR109A expression in mammary gland tissue through conventional western blot techniques to support the findings.

A: Thank you very much for your comments. The concentration of GPR109A in immunohistochemistry was 1:50 (line 82). I've added this description to the material method. I've also added brief description of experimental procedure for reproducibility in the result part. We also used WB technology to detect the protein level of GPR109a in two groups of mammary gland. We can find that the protein levels of GPR109A in mastitis group is significantly higher than that of Control group (Fig 2).

Fig 2. Protein level of GPR109A in mammary gland. (a, b) Protein levels of GPR109A. The values are presented as the means ±SD (∗p<0.05 and ∗∗p<0.01).

  1. Page 8, Activation of GPR109A.............signaling pathways in BMECs- The results in Fig. 3a GPR109A showed overexpression induced by niacin. The expression of housekeeping protein b-tubulin is not well convincing. It is a general notion that housekeeping protein expression should have clear band intensity. This reviewer suggests showing better quality b-tubulin bands in western blot. (Authors may consider the data presented in the reference given here in Fig3 ref. Sun J, et. al. Int J Mol Sci. 2017; 18(12): 2621. Published 2017 Dec 5. Doi: 10.3390/ijms18122621) from same group.

A: Thank you very much for your comments. I've replaced a clearer β-tubulin bands in the paper (Fig 3).

Fig 3 Niacin can activate GPR109A/NRF2/autophagy signal pathway. Niacin can activate GPR109A/NRF-2/autophagy signal pathway. The cells were collected at 0 h, 3 h, 6 h, 12 h and 24 h to extract the total protein. The total protein was prepared and subjected to western blotting using GPR109A, NRF-2, HO-1 and β-tubulin antibodies. T-NRF-2 means total NRF-2. (a-d) The protein levels of GPR109A, NRF-2 and HO-1. The cells from different experimental groups were treated with niacin or shRNA+niacin for 24 h, and then, the total protein was collected. N-NRF-2 means NRF-2 in the nucleus. C-NRF-2 means NRF-2 in the cytoplasm. (e-h) The protein levels of GPR109A, C-NRF-2, N-NRF-2 and HO-1. Each immunoreactive band was digitized and expressed as a ratio of the β-tubulin level. (i) The immunofluorescence results of the assay for NRF-2. (j) The relative fluorescence intensity of NRF-2/ARE. The mRNA levels were determined by qRT-PCR. (k) The mRNA levels of ATG12, ATG4D, p62, ATG5, ULK1, ATG4B, Beclin, LC3B and ATG7 were normalized to the level of β- actin. The values are presented as the means ±SD (∗p<0.05 and ∗∗p<0.01).

Authors just changed the b-tubulin band. Authors failed to do this experiments. Because, they have to repeat all other protein mentioned in Fig. 3a should be based on the b-tubulin blot. Not a piece of band from other WB. This is not acceptable.

Figure 3e. The NRF2 expression in nuclear and cytosolic fraction well characterized. However, authors fail to describe how they separate nuclear and cytosolic fractions. It is neither described in materials and methods nor cited any references. (Couple of reference given here authors attention Antalis, T.M et. al Nucleic Acids Res. 1991, 19, 4301, and Giri et. al Int. J. Mol. Sci. 2019, 20(18), 4559).

A: Thank you very much for your comments. I have added the experimental method of cytoplasm and nucleus separation to the material method in the paper (line 123- 127).

Separation of nucleus and cytoplasm

We inoculated the BMECs into the cell culture dish. When the cells reached about 80%, GPR109A-shRNA, niacin or LPS were used to treat the BMECs. Cells were washed twice with cold PBS after the above treatment and subjected to nuclear and cytosolic fractionations with modification described by Antalis et al and Giri et. al.

Authors need to perform the experiments not just cite the references. The data presented are old data not the new one following this cited references.

Figure 3i. The immunofluorescence pictures resolutions are poorly presented. It should be showed in higher magnification with clear vision of cell structure. This reviewer does not recommend this figure for publications.

A: Thank you very much for your comments. I re-import Figure 3i. Now when you zoom in, you can see the cell structure clearly (Fig 4).

Fig 4. Immunofluorescence image of Nrf2 in nucleus. Immunofluorescence results of Nrf2.

Figure 3i is not convincing at all even after zooming it. This figure needs to have clear cell structure visible. This figure is not acceptable.

Figure 5 e-m. Page number 10. The authors showed the transcripts level of ATG4D, Beclin, ATG12, ATG5 and LC3B (Fig 5e, f, g, j, k, l). The level of ULK1 is not sufficient to claim autophagy. Authors need to show and protein level as well.

A: The questions you ask are crucial. Because Nrf2 is a transcription factor, it can combine with DNA in the nucleus to promote the transcription of downstream target genes. In our study, niacin can significantly promote Nrf2 nuclear import, so we want to know whether Nrf2 can also promote the expression of downstream autophagy related genes in the nucleus. So, we do qRT-PCR mainly to clarify that Nrf2 can regulate the expression of autophagy related genes at the transcription level. However, the specific pathway of niacin activating downstream autophagy has not been studied. But in this study, we can clearly see the formation of autophagy lysosomes in the electron microscopy results. Western blotting of LC3B and P62 also showed that

niacin can activate autophagy. Therefore, our next plan is to elucidate the pathway of niacin activating autophagy. And we will continue to consider publishing our results in IJMS. Thank you again for your valuable comments.

If this is the explanation, then authors need not to claim autophagy phenomena but may mention as a speculative observation.

Specific Minor Comments:

Authors should revise, as there are missing spaces, abbreviations (NT), superscripts, and italics. Few of missing spaces are given below.

Introduction section-
Page 1, Line 5, spaces in periods “$35 billion annually .”
A: Thank you very much for your comments. I have revised in the paper. Page 2, Line 31, spaces in periods “glycometabolism and inflammation . ” A: Thank you very much for your comments. I have revised in the paper. Page 2, Line 32, spaces in periods “ inhibiting cell survival .”
A: Thank you very much for your comments. I have revised in the paper. Materials and Methods section
Page 3, Line 15, spaces in words “ofacells .”
A: Thank you very much for your comments. I have revised in the paper. Page 3, Line 15, correct this “was 0.3 × 105.”
A: Thank you very much for your comments. I have revised in the paper. Page 6, Line 32, convert it to italics “ in vivo .”
A: Thank you very much for your comments. I have revised in the paper.

Results Sections-

Page 8, Line 19, spaces in periods “autophagy-related genes .”

A: Thank you very much for your comments. I have revised in the paper.

Page 9, Line 18- 19, Need references “Previous studies have shown that the ratio of (AMP+ADP)/ATP is the key signal for activating AMPK .”

A: Thank you very much for your comments. I have revised in the paper.

Discussion sections-

Page 12, Line 20, spaces in periods “mediators in adipocytes or monocytes . ”

A: Thank you very much for your comments. I have revised in the paper.

Page 13, Line 5, spaces in periods “renewal of some organelles . ”

A: Thank you very much for your comments. I have revised in the paper.

Page 13, Line 8, spaces in periods “injury in mice . ”

A: Thank you very much for your comments. I have revised in the paper.

Page 13, Line 14, spaces in periods “promotes NRF-2 nuclear import . ”

A: Thank you very much for your comments. I have revised in the paper.

Once these issues are addressed, then I would happy to recommend this manuscript for publications.

Author Response

Reviewer 2

Thank you very much for taking the time to review my article. Your questions are very professional. If there is anything wrong with my answer, please contact me in time. Thank you again for your comments.

  1. Where is the catalogue number?

A: Thank you very much for your valuable comments. Because the sequence of the plasmid was designed and synthesized in GenePharma company, only the product number of the plasmid was used as shown.

Gene name:  LOC527744(GPR109A)Lot No:      A8215

Vector name:       pGPU6/Neo CAT:      C02003

shRNA template sequence:

Sequence structure of transcripts

Sequencing results

The sequencing results are obtained by forward primer sequencing, and the positive sense strand can be directly found in the sequencing results

  1. Hengyuan company name is not available. Do you have contact information of this company?

A: Thank you for your comments. This is the official website of their company. http://www.shhyswsj.com/hyshsj2011-Products-27956286/

http://www.shhyswsj.com/hyshsj2011-Products-27601301/

http://www.shhyswsj.com/hyshsj2011-Products-28886992/

  1. What is the mobile phase used for separation of ATP, ADP, and AMP?

A: Thank you for your comments. My mobile phase used for separation of ATP, ADP, and AMP is NaH2PO4 (50mmol/L, 1% methanol) (line160-161).

  1. Authors just changed the b-tubulin band. Authors failed to do this experiments. Because, they have to repeat all other protein mentioned in Fig. 3a should be based on the b-tubulin blot. Not a piece of band from other WB. This is not acceptable.

A: Thank you for your valuable advice. I have carried out WB experiments on all proteins.

  1. Authors need to perform the experiments not just cite the references. The data presented are old data not the new one following this cited references.

A: Thank you for your valuable comments. I've made changes in the paper(Line 126-127).

We inoculated the BMECs into the cell culture dish. When the cells reached about 80%, GPR109A-shRNA, niacin or LPS were used to treat the BMECs. Cells were washed twice with cold PBS after the above treatment[29,30] and subjected to nuclear and cytosolic fractionations according to the Nuclear and Cytoplasmic Protein Extraction Kit (Cat: P0027, beyotime, Shanghai,China).

  1. Figure 3i is not convincing at all even after zooming it. This figure needs to have clear cell structure visible. This figure is not acceptable.

A: Thank you for your valuable comments. I did the immunofluorescence test of Nrf2 again.

  1. If this is the explanation, then authors need not to claim autophagy phenomena but may mention as a speculative observation.

A: Your suggestions are very valuable. I have changed this description in this paper (line 377-378, 472-473)

377-378 These results indicated that autophagy could be observed after GPR109A is activated, which may be related to the anti-inflammatory effect of GPR109A.

472-473 Therefore, we speculate that GPR109A may also play an anti-inflammatory role by activating autophagy.

Round 3

Reviewer 2 Report

This revised version is much more improved. Experimental data are presented clearly now. However, I have few comment and concerns that given below need to addressed before acceptance.

Comment:

The term autophagy should be removed from the title before acceptance because authors agreed that autophagy is a speculative based on their preliminary findings.

Authors quoted as 

" A: Your suggestions are very valuable. I have changed this description in this paper (line 377-378, 472-473)

377-378 These results indicated that autophagy could be observed after GPR109A is activated, which may be related to the anti-inflammatory effect of GPR109A.

472-473 Therefore, we speculate that GPR109A may also play an anti-inflammatory role by activating autophagy." in response to this reviewer comment number 7.

Concerns:

I have one concerns on experimental performance.  Because, I have submitted comments on 17th April (literally 18th April at China time)  and revised version came back on April 22, 9.00PM. Basically authors got four days in hand to start and finish the experiments.

My concerns is how come the authors able to do all three western blot presented in figure 1a. and immunofluorescence result in Figure 1i just in four days. I understand authors perform all the experiments in same time. However, usually in vitro cell culture need at least 24-48 hr for cells to grow  followed by treatment for another 24 hr maximum time point as described. Furthermore, incubation of antibody took overnight for fluorescence study as described in methods Line 195. Almost four days gone. I am curious how they finish all the experiment just in four days unless they have performed the experiments previously.

Author Response

1.The term autophagy should be removed from the title before acceptance because authors agreed that autophagy is a speculative based on their preliminary findings.

A:Thank you for your advice. I have revised the title.

2. I have one concerns on experimental performance. Because, I have submitted comments on 17th April (literally 18th April at China time) and revised version came back on April 22, 9.00PM. Basically authors got four days in hand to start and finish the experiments.

My concerns is how come the authors able to do all three western blot presented in figure 1a. and immunofluorescence result in Figure 1i just in four days. I understand authors perform all the experiments in same time. However, usually in vitro cell culture need at least 24-48 hr for cells to grow  followed by treatment for another 24 hr maximum time point as described. Furthermore, incubation of antibody took overnight for fluorescence study as described in methods Line 195. Almost four days gone. I am curious how they finish all the experiment just in four days unless they have performed the experiments previously.

A: Thank you for your question. I'm very sorry for the trouble. I should explain it in my last reply. In fact, at the time of the first revision, we had collected the samples of this part of the experiment, but I only ran the β-tubulin alone. And then you asked this question again, so we ran all the proteins from the samples collected in the last revision, including the β-tubulin. As for immunofluorescence, it has been stained with red and green respectively before, but considering that there are many cells in the red fluorescence experiment, it is more representative, so we only showed the red fluorescence. In fact, for the first time you raised the issue of immunofluorescence, I thought there was a problem with the quality of the picture. So we just enhanced the image quality, and later realized that you might want to have clearer cell images.